# Zibibbo Grape Seeds’ Polyphenolic Profile: Effects on Bone Turnover and Metabolism

**DOI:** 10.3390/ph17111418

**Published:** 2024-10-23

**Authors:** Mariagiovanna Settino, Samantha Maurotti, Luca Tirinato, Simona Greco, Anna Rita Coppoletta, Antonio Cardamone, Vincenzo Musolino, Tiziana Montalcini, Arturo Pujia, Rosario Mare

**Affiliations:** 1Department of Clinical and Experimental Medicine, University “Magna Græcia” of Catanzaro, 88100 Catanzaro, Italy; margiov.settino@gmail.com (M.S.); smaurotti@unicz.it (S.M.); tmontalcini@unicz.it (T.M.); 2Department of Medical and Surgical Sciences, University “Magna Græcia” of Catanzaro, 88100 Catanzaro, Italy; tirinato@unicz.it (L.T.); simona.greco001@studenti.unicz.it (S.G.); 3Pharmacology Laboratory, Department of Health Sciences, Institute of Research for Food Safety and Health IRC-FSH, University “Magna Græcia” of Catanzaro, 88100 Catanzaro, Italy; annarita.coppoletta1@gmail.com (A.R.C.); tony.c@outlook.it (A.C.); 4Laboratory of Pharmaceutical Biology, Department of Health Sciences, Institute of Research for Food Safety & Health IRC-FSH, University “Magna Græcia” of Catanzaro, 88100 Catanzaro, Italy; v.musolino@unicz.it; 5Research Center for the Prevention and Treatment of Metabolic Diseases, University “Magna Græcia” of Catanzaro, 88100 Catanzaro, Italy

**Keywords:** grape seeds, polyphenols, flavonoids, osteoporosis, bone health

## Abstract

Background: The consumption of seeds as food has become increasingly common due to their numerous health benefits. Among these, the seeds of the Zibibbo grape from Pantelleria, a native species of southern Italy, remain largely unexplored and are usually considered waste material from viticulture. Nevertheless, Zibibbo grape seeds may offer health benefits, particularly for the elderly and people with metabolic disorders, due to their potential content of beneficial compounds such as polyphenols. Methods: The Zibibbo grape seeds extract (ZSE) was characterized using UV-visible spectrophotometry and HPLC chromatography. The antioxidant activity of ZSE was measured by different colorimetric assays and Electronic Paramagnetic Resonance (EPR). Additionally, specific in vitro tests were conducted on human osteoblast cell lines (Saos-2 and MG63) aiming to evaluate the ZSE’s effects on bone turnover and metabolism. Western blotting was used to assess the impact on specific proteins and pathways related to bone health. Results: The ZSE contained almost ~3 mg/mL of carbohydrates and phenolic compounds, including rutin (~6.4 ppm) and hesperidin (~44.6 ppm). The extracts exhibited an antioxidant activity greater than 90% across all tests performed. Moreover, the Zibibbo seed extracts exerted a significant proliferative effect on the Saos-2 cell human osteoblast-like cell line, also modulating the phosphorylation of specific kinases involved in cell health and metabolism. Conclusions: Zibibbo grape seeds are rich in phenolic compounds, especially flavonoids with strong antioxidant and free radical scavenging properties. ZSE demonstrated beneficial effects on bone metabolism and osteoblast proliferation, suggesting potential for counteracting osteodegenerative conditions like osteoporosis. If confirmed through further studies, Zibibbo grape seed phenolic compounds could serve as an adjunctive therapy for osteoporosis, helping to slow aging and bone degeneration.

## 1. Introduction

Following the Mediterranean diet on a daily basis is commonly associated with different benefits to health and pathological status [1]. Among vegetables and plant derivatives, the consumption of fresh or dried seeds is becoming widespread because of the macromolecules they contain. In fact, seeds are usually considered a waste material from fruit processing procedures, but several studies have shown that they contain multiple beneficial molecules such as vitamins, organic acids and polyphenolic compounds, including several flavonoids, which could be useful in various pathological conditions [2]. In particular, several beneficial effects have recently been associated with the consumption of grape seeds and their derivatives, which have been shown to be advantageous in improving gastrointestinal mobility [3], cardiovascular and metabolic diseases [4,5] and even in cancer therapy [6]. Despite the growing interest in grape seeds, there is still a lack of comprehensive studies on certain varieties of grapes, particularly regarding their biochemical diversity and potential health applications. It is well known that the biochemical properties of grapes can vary significantly depending on the botanical species and the climatic conditions [7]. This variability suggests the importance of investigating specific grape varieties to better understand their unique compositions and potential health benefits.

Zibibbo grape is a peculiar grape species, cultivated in some Italian regions and predominant on Pantelleria island [8]. Despite the interest of the scientific community in grape seeds, the Zibibbo grape was mostly, if not exclusively, studied for winemaking [9] or for making raisins [10]. Therefore, the only studies available in the literature concern “passito” wines and the microbiological characteristics of grapes for producing them [11]. As a result, there is a significant gap in the literature regarding the potential health benefits of Zibibbo seeds. In fact, within the literature there is a lack of information about their potential nutritional or medical applications. However, recent scientific evidence demonstrates that grape seeds are also rich in polyphenolic and antioxidant substances, which could play a crucial role in various pathologies related to oxidative stress [12]. In addition, the consumption of seeds has been shown to help mitigate osteodegenerative diseases such as osteoporosis, a frequent condition that occurs in post-menopausal women and during the physiological aging process [13].

This study aims to fill the gap in the literature about Zibibbo seeds by characterizing their bioactive profile, focusing particularly on the polyphenolic compounds and antioxidant properties they contain. Additionally, this study offers an evaluation of antioxidant power using different analytical methods, finally investigating the biocompatibility and safety of Zibibbo seed extracts (ZSEs) on osteoblast-like cell lines, with the goal of exploring their potential benefits through in vitro models. By addressing this knowledge gap, we aim to contribute to the understanding of the unique properties of Zibibbo grape seeds and clarify their potential beneficial effects on bone tissues and the aging processes.

## 2. Results

### 2.1. Chemical Analyses

Purified ZSEs were analyzed using an UV-Vis spectrophotometer after being diluted in a 1:200 ratio with the same hydro-alcoholic mixture used for the extraction. All ZSEs showed a main and characteristic peak between 250 nm and 300 nm, with maximum absorbance at λ~280 nm (Figure 1).

Zibibbo seeds extracts were characterized through different analytical approaches. The results for the main macromolecules detected and quantified in the ZSEs are listed and described in Table 1. In detail, according to the Bradford assay, the protein content was ~0.69 ± 0.02 mg/mL, equivalent to that in bovine serum albumin (EBSA—Table 1). The amounts of carbohydrates contained in the ZSEs were estimated using the sulfuric acid–phenol assay and compared with a stock mixture containing glucose, sucrose and fructose in equal ratio (conc. 6.25–75 ppm). The mean carbohydrates content in the ZSEs was ~3.23 ± 0.49 mg/mL, equivalent to that in sucrose (SEq—Table 1).

A total phenolic content assay provided information about the presence of phenolic compounds and tannin-like molecules in the ZSEs. The reaction with Folin–Ciocalteu’s reagent allowed us to detect ~1.575 ± 0.19 mg/mL molecules, equivalent to gallic acid (GAE—Table 1). The application of the sodium nitrite–aluminum chloride assay allowed us to discriminate flavonoids. All samples demonstrated the peculiar and characteristic red color of molecules with a catechinic structure. The spectrophotometric analyses showed a content of ~2.05 ± 0.25 mg/mL, and mean flavonoids content was ~102 ± 12 mg for every gram of seed powder investigated (Table 1).

HPLC analyses confirmed the presence of flavonoids in the extracts and specifically detected and quantified them as a function of the standard molecules available (Figure 2). In detail, the injection peak, which is appreciable in all HPLC analyses, was detectable after ~1 min retention. Rutin and hesperidin signals eluted, respectively, with an R.T. of 7.5 min. and 12.5 min., with the latter having a concentration approximately seven times higher than rutin. In fact, the mean concentrations of rutin and hesperidin in ZSE were approximately ~6.4 ± 0.3 ppm and 44.6 ± 4.5 ppm (Figure 2). Moreover, The HPLC analyses highlighted the presence of a large peak (R.T. ~17 min) attributable to another polyphenolic substance, which is presumably a flavonoid but remains unknown. The increase in absorbance detected after ~23 min is due to the increase in the organic phase in the elution mixture, but it does not represent a peak (Figure 2).

The Zibibbo seed extracts showed strong antioxidant and free-radical scavenging activities, regardless of the technique used (Figure 3). In detail, the interaction with the 2,2-Diphenyl-1-picrylhydrazyl (DPPH) was expressed as percentage of free radicals inhibited (%I). In this case, the ZSEs showed a percentage of antioxidant activity ~82 ± 3% in comparison with a solution of L-ascorbic acid (5 mg/mL) enrolled as a positive control (Figure 3A). 

The ABTS assay echoed the DPPH colorimetric assay’s results, but provided superior data, finally confirming the ZSEs’ antioxidant activity of ~97 ± 1% I (Figure 3B). The use of Electronic Paramagnetic Resonance (EPR) allowed us to assess the free radical scavenging activity of ZSE. L-ascorbic acid (5 mg/mL solution) was also enrolled as positive control in this test. The pure DPPH solution had a spectral area (a.u.) of ʃ = 1754 ± 3. The mixture obtained by adding ascorbic acid had a spectral area of ʃ = 42 ± 2. However, the addition of the Zibibbo seeds extract to the DPPH solution showed spectral area values (ʃ) between 56 ± 3 and 93 ± 3, finally demonstrating an average free-radical scavenging activity of 95.75% (Figure 3C).

### 2.2. Effect of ZSE on Saos-2 Osteoblast In Vitro Model

We investigated the effects of Zibibbo seed polyphenols on Saos-2 osteoblast cells. Specifically, changes in cell proliferation, ERK1/2 and AKT proteins phosphorylation were examined, as well as the intracellular production of oxygen free radicals, aiming to evaluate ZSEs’ influences on bone turnover and metabolism. After 24 h of incubation, the ZSEs showed a statistically significant proliferative effect on a Saos-2 cell line at doses between 5 and 40 µg/mL as well as at a dose of 0.31 µg/mL compared to the control group (Figure 4A). In addition to this, in the Saos-2 cell line, the ZSEs showed a statistically significant proliferative effect at doses greater than 1.25 µg/mL compared to the control group treated with ethanol 70%, finally showing a dose-dependent proliferative effect (Figure 4A).

The effect of the ZSEs on ERK1/2 and AKT phosphorylation was assessed by treating Saos-2 cells with a 10 nM dexamethasone medium containing the ZSEs in concentrations between 2.5 µg/mL and 40 µg/mL (Figure 4B,C). The incubation with ZSEs increased ERK1/2 phosphorylation in a dose-dependent manner compared to the untreated control group (*p* = 0.0011; Figure 4B). In detail, treatment with the ZSEs resulted in a statistically significant increase in p-ERK1/2 at doses of 10 µg/mL, 20 µg/mL (*p* < 0.001) and 40 µg/mL (*p* < 0.01) (Figure 4B). Additionally, on the same cell line, the incubation with ZSE also increased AKT phosphorylation in a dose-dependent manner, compared to the untreated control group (*p* < 0.0001), thus resulting in a statistically significant increase in AKT phosphorylation at doses of 10 µg/mL (*p* < 0.05), 20 µg/mL and 40 µg/mL (*p* < 0.0001) (Figure 4C).

Moreover, treatment with ZSEs was shown to significantly reduce the amount of reactive oxygen species (ROS) released from cells regardless of the dose examined; finally, a dose-dependent trend was also shown for this parameter (Figure 4D).

### 2.3. Effect of ZSE on MG63 Osteoblast In Vitro Model

Given that the MG63 cell line is an advantageous model for investigating the early stages of osteoblastic differentiation, we used this cell line to test the ZSEs using same conditions and dosages previously employed for the Saos-2 cells [14].

After 24 h of incubation with MG63 cells, treatment with 10 µg/mL of ZSE showed a significant increase in cell viability compared to the control group and the ethanol-treated control (Figure 5A, *p* < 0.05). On the contrary, the 40 µg/mL dose resulted in a significant reduction in cell viability (Figure 5A, *p* < 0.05). Similarly, the evaluation of pERK1/2 protein expression showed a reduction in the phosphorylated form of the protein at the 10 µg/mL dose, but did not reveal any significant changes at the other doses investigated. On the contrary, treatment with the ZSEs increased AKT phosphorylation in a dose-dependent manner, if compared to the untreated control group (Figure 5C, *p* < 0.0001). This resulted in a statistically significant increase in protein phosphorylation at doses of 10 µg/mL and 40 µg/mL (*p* < 0.001) as well as treatment with 20 µg/mL (*p* < 0.01) (Figure 5C).

Moreover, and similarly to the Saos-2 cells, treatments with ZSEs were able to significantly reduce the number of reactive oxygen species (ROS) released from cells, regardless of the doses examined, and a dose-dependent trend was also shown for this parameter. (Figure 5D, *p* < 0.0001 in all cases).

## 3. Discussion

Adherence to the Mediterranean diet appears to be the most beneficial approach to pursuing healthy aging. Most of the recognized benefits of the Mediterranean diet are related to the naturally high contents of polyunsaturated fatty acids, vitamins and polyphenolic compounds in the foods. In particular, polyphenols were demonstrated to play a key role in promoting people’s health and the prevention of various diseases, such as type 2 diabetes, cardiovascular diseases, obesity and even neurological pathologies. In fact, polyphenols are known to be directly involved in glucose regulation and transportation, as well as in the protection of β-cells from glucose toxicity [15,16]. They also exerted effects on reducing platelet activation and reducing LDL cholesterol [17], on adipocyte oxidation [18] and on psychological and cognitive health, and there were cases in which the consumption of coffee, tea and red wine resulted in a reduction in depressive symptoms and perceived stress [19].

Seeds have long played crucial roles in human nutrition throughout history, but in recent decades, they have captured the attention of the scientific community because of the potential health benefits associated with their regular consumption. In particular, different nutrients, fat-soluble bioactives and polyphenols have been abundantly found in seeds collected from hemp [13], black cumin, chia, flax, perilla, pumpkin, quinoa and sesame, thus conferring them different health benefits that are mainly related to their antioxidant activity, good bioavailability and toxicological safety [20].

Although several research groups have analyzed grape seeds, most of them focused on lipophilic molecules [21].

The Zibibbo seed extracts were analyzed unconditionally and regardless of all the information in the literature concerning grape berries and juice. This approach was necessary and justified by the qualitative and quantitative differences existing between the molecules contained in leaves, fruits and seeds belonging to the same plant species [22]. These differences, in agreement with data described in the literature, did not facilitate the use of a predictive approach regarding the molecules contained in the extract, especially considering this is the first study about Zibibbo seeds available in the literature. The extraction procedure was effective in isolating the hydrophilic compounds present in the Zibibbo seeds, with the extracts containing a substantial number of phenolic compounds besides a small fraction of proteins and carbohydrates. Interestingly, the total carbohydrates content was analogous to the overall phenolic content detected in the extract. This phenomenon may be justified by the fact that most of the phenolic compounds in fruits and vegetables are spontaneously conjugated to a glucidic molecule, particularly D-glucopyranose or rutinose. In agreement with our hypothesis, several studies in the literature confirm that phenolic compounds are predominantly in the glycosylated form [23].

The phenolic fraction was rich in flavonoids and gallic acid-like polyphenols, which overall showed a remarkable antioxidant capacity and strong scavenging activity against free radicals with all in vitro assay proposed in the study. This aligns with findings from the current literature, supporting the notion that the antioxidant and free radical scavenging effects can be attributed to the phenolic compounds, placing specific emphasis on the flavonoids contained within the Zibibbo seed extract (ZSE) [24,25].

The use of the HPLC apparatus allowed the analytical discrimination and quantification of rutin and hesperidin contained in the extracts. Interestingly, one of the phenolic compounds contained in the extract was unidentified due to the absence of the reference standard molecule. Nevertheless, the use of chromatography allowed us to gain more detailed information about the analytical composition of the extract compared to the exclusive use of spectrophotometric analyses, as frequently reported in the literature [26,27]. This represents an advantage of the data described in this study, given the concomitant presence of colorimetric assays aimed at completing the chemical–physical characterization of the extracts.

The chemical composition of the molecules found in the Zibibbo grape seeds is similar to other grape species previously analyzed, and data available in the literature include slight differences related to the diversity between plant species investigated as well as by the differences in cultivation conditions [21,28].

The abundances of the compounds mentioned above in the extract contribute to its efficacy in reducing oxidative stress and neutralizing free radicals, as substantiated by established scientific knowledge [29]. The extracts we obtained from Zibibbo seeds, likely because of their abundance in phenolic compounds, showed a sustained antioxidant effect in vitro and proved to be effective in reducing the reactive oxygen species produced by both the MG63 and Saos-2 cell lines. The oxidative phenomena and the production of free radicals are involved in the aging process and in the onset and worsening of several pathologies including osteoporosis; therefore, the flavonoids contained in Zibibbo seeds could help to contrast these pathological conditions. In support of our hypothesis, ZSE also showed a proliferative effect on Saos-2 osteoblast-like cells and, partially on the MG63 cell line. The difference observed in the cells’ proliferation could be due to the different stages of differentiation in the cell lines. In fact, Saos-2 cells are a useful model for studying the later stages of osteoblast differentiation in human cells, while MG63 cells correspond to the early stages [30].

The trends observed in the cell viability tests are consistent with the results obtained for the proteins and pathways we analyzed in both cell lines. In fact, phosphorylated ERK and AKT are protein kinases playing a crucial role in promoting cells’ survival and proliferation, also inducing the expression of genes involved in DNA replication and cell division. The results obtained on early differentiated MG63 cells and those made by the osteoblast cell line Saos-2, both suggest the effectiveness of ZSE in promoting bone turnover and opposing to the onset of osteoporosis, a pathological condition typically characterized by a specific reduction in osteoblastic cells in bone tissue [31,32]. The promising results obtained with in vitro cellular models and the antioxidant power demonstrated in laboratory tests suggest that the consumption of Zibibbo grape seeds may contribute to reducing osteodegenerative processes and the onset of osteoporosis.

This study analyzes, for the first time, the effect of flavonoids contained in Zibibbo grape seeds, finally suggesting their direct involvement in bone turnover. These represent the major advantages and innovations of the results obtained, especially considering that most of the studies in the literature are exclusively based on molecules contained in fruits and pulp, such as anthocyanins [33,34].

## 4. Materials and Methods

### 4.1. Chemicals and Reagents

Dexamethasone (DEX) and Bradford reagent for protein quantification, as well as Folin–Ciocalteu reagent, phenol, sodium carbonate, gallic acid, 2,2-diphenyl-1-picrylhydrazyl (DPPH), aluminum chloride, sodium nitrite and sodium hydroxide required for spectrophotometric assays, were purchased from Sigma-Aldrich (St. Louis, MO, USA). Gibco (Life technologies, Grand Island, NY, USA) provided Dulbecco Modified Eagle Medium (DMEM) and McCoy’s 5A modified medium, fetal bovine serum (FBS), and trypsin-EDTA; penicillin/streptomycin was purchased from Lonza (Basel, Switzerland). The cellular ROS Assay Kit [AB113851] DCFDA/H2DCFDA was provided by Abcam (Abcam Limited, Cambridge, UK). Methanol and all other solvents and reagents were of analytical grade (Carlo Erba, Milan, Italy).

### 4.2. Seeds Supply, Processing and Extraction

The seeds were obtained from the Zibibbo grape variety, which was purchased from company Agricultural Cooperative Capers Producers SOC. COOP. The company is located in Pantelleria (TP) – Sicily, a volcanic island in the Strait of Sicily characterized by a subtropical Mediterranean climate, with hot, dry and humid summers, as well as mild, frost-free winters. The island benefits from constant sea breezes that moderate daytime summer temperatures, resulting in a relatively stable thermal environment.

The Zibibbo berries were dried naturally by exposing them to sunlight and, once they arrived in the laboratory, each grape was individually opened, and the seeds removed with a scalpel and tweezers. The seeds were washed and dried to deprive them of the adherent pulp, and finally stored in a −80 °C freezer until the next processing step. The grape seeds were crushed manually in a mortar, using liquid nitrogen to ease the process and avoid temperature increases due to friction. The powder thus obtained was subsequently used for the extraction procedure, as previously described by Spigno et al., with slight modifications [35]. In detail, the seed powder was incubated with a hydro-alcoholic solution of ethanol and milliQ^®^ water (Carlo Erba, Milan, Italy) in a 7:3 molar ratio. The ratio between the seed powder and extractive medium was 1:50 weight/volume. The mixture was incubated for 24 h under continuous stirring and a temperature set at 45 °C. 

The extract obtained was subsequently purified by two different centrifugation cycles. The first cycle was performed using an Eppendorf 5810R centrifuge set at 2500 rpm and 18 °C for 5 min, while the second one had a duration of 8 min during which the sample was centrifuged at 14,000× *g* by a ThermoFisher Scientific – Fresco 21 apparatus (ThermoFisher Scientific, Rosano, MI, Italy), with the temperature set at 4 °C. Pellets were always disposed of as waste material, while the supernatant was further characterized.

### 4.3. Chemical Analyses

Zibbibo seed extracts (ZSEs) were analyzed with ThermoScientific-Genesys 150^®^ (ThermoFisher Scientific, Rosano, MI, Italy) before being characterized. All analyses were performed using quartz cuvettes. The instrument was set to scansion mode with a wavelength (λ) range set between 200 nm and 700 nm. The hydro-alcoholic mixture used for extraction procedure was also used as blank and diluent before analyses. ZSE were characterized in terms of the content of proteins, carbohydrates and polyphenols as previously described [13]. In addition to this, we also specifically determined flavonoids contained in the extracts according to a method recently published in the literature [36]. 

The proteins contained in the ZSEs were quantified using the Bradford solution. In detail, Bradford solution was mixed with milliQ^®^ water in a 1:4 ratio. Then, 2 µL samples were added to the mixture and the absorbance was read at λ = 595 nm. Bovine serum albumin (BSA) with a fixed concentration was used as a standard reference.

The carbohydrate profile of the ZSE was assessed using the phenol-sulfuric acid assay. Briefly, the phenol-sulfuric acid assay involves the incubation of 500 µL of sample with 1.25 mL of concentrated sulfuric acid. After vortexing, 250 µL of an aqueous solution containing phenols (5% *w*/*v*) was added to the sample. The obtained mixture was incubated on ice for 20 min and was finally read using a UV-vis spectrophotometer set at λ = 490 nm. A properly diluted stock solution containing sucrose (conc. 6.25–75 ppm) was used as reference standard. 

The total content of polyphenolic molecules (TPC) with a tannin-like structure was evaluated by incubating the sample for 5 min with Folin–Ciocalteu’s reagent in a 1:5 ratio. Subsequently, a solution of sodium carbonate was added to the mixture and the absorbance was read at λ = 760 nm. Gallic acid was used as reference standard molecule.

### 4.4. Evaluation of Antioxidant and Free Radicals Scavenging Activities

The ZSEs were also characterized in terms of antioxidant activity and free radicals scavenging activity using three different approaches. The free radical scavenging activity of the ZSEs was evaluated by incubating the sample with two free radicals, 2,2-Diphenyl-1-picrylhydrazyl (DPPH) and 2,2′-Azino-bis-3-ethylbenzothiazoline-6-sulfonic acid (ABTS), as previously described in the literature [37]. 

For the DPPH assay, 25 µL of sample were incubated in the dark for 30 min with 0.004% *w/v* radical solution and absorbance detected at λ = 517 nm. 

Concerning the ABTS assay, a 2.45 mM solution of potassium persulfate was previously prepared and mixed with a 7 mM aqueous solution of ABTS, followed by an incubation overnight protected from light, with the aim of producing the ABTS radicals. The working solution was obtained by adding milli Q^®^ water at a volume equal to 35 times the ABTS radical mixture. Finally, an appropriately diluted ZSE was mixed with the working solution in a 1:4 ratio and the absorbance was measured at 734 nm after 5 min incubation. In both assays, the variation in radical concentration was detected and quantified by a UV-Vis spectrophotometer, and the percentage of inhibition was calculated using the following equation:I (%) = [(A_0_ − A_1_)/A_0_] × 100

A_0_ = absorbance of negative control; A_1_ = absorbance of extracts/standards.

The scavenging activity of the ZSE against DPPH radicals was also evaluated through Electronic Paramagnetic Resonance (EPR). The decrease in EPR signal intensity (spectral area) represented the free radical scavenging activity of the extracts. The scavenging activity was finally expressed as the percentage of reduction in curve area ± standard deviation (a.u. − ʃ ± S.D.) Ascorbic acid was used as the reference molecule (5 mg/mL).

### 4.5. HPLC Analyses

The HPLC apparatus consisted of a ThermoFisher Scientific Vanquish System Base VC-S01-A-02 equipped with a VC-P20-A-01 quaternary pump, a VC-A13-A-02 split sampler, a VC-C10-A-03 column compartment and a VF-D40-A UV/VIS variable wavelength detector (ThermoFisher Scientific, Rosano, MI, Italy). The Chromeleon^®^ software version 7.2 was used for data processing and was purchased from ThermoFisher Scientific (Rosano, MI, Italy). The analytical column was an Acclaim^®^ 120 reverse phase C18 (100 mm 4.6 mm), with a 5 µm particle size. The mobile phase was a H_3_PO_4_ 10 mM aqueous solution/ACN with different ratios during the analysis. The column temperature was 30 °C and the absorbance was acquired at wavelengths of 260 nm and 270 nm. Peak identification was carried out through a comparison of the retention times with those obtained with key molecules representing the main categories of flavonoids (Appendix A). In detail, rutin was used as a model flavanol bearing a catechol ring; hesperidin was used as a flavanone containing a methoxy derivative; and naringin and apigenin were recruited as flavones containing p-hydroxy-phenolic rings in CIS and TRANS positions, respectively (Appendix A). Suitable calibration curves (min conc. 15.625 ppm and max conc. 250 ppm–r^2^ > 0.97 in all cases) were used for the identification of flavonoids contained in samples. The samples (20 µL) were injected into HPLC apparatus with the total acquisition time set at 25 min.

### 4.6. Cell Line and Culture Conditions

The Saos-2 and MG63 cell lines were obtained from ATCC (ATCC, Milan, Italy) and were used in this study, respectively, as human osteoblast-like and fibroblast cells lines. Saos-2 cells were maintained in McCoy’s 5A (Gibco, Carlsbad, CA, USA) supplemented with 15% fetal bovine serum (Gibco, Carlsbad, CA, USA), while the MG63 cell lines were maintained with DMEM 1×-high glucose (Gibco, Carlsbad, CA, USA) supplemented with 10% fetal bovine serum (Gibco, Carlsbad, CA, USA). Both media were also supplemented with a 1% penicillin–streptomycin solution (PAA, Linz, Austria), to avoid bacteria proliferation. Maintenance conditions were temperature set at 37 °C with 5% CO_2_. All cell cultures were harvested using trypsinization and were subcultured twice a week, and both were incubated with dexamethasone 10 nM (Sigma Aldrich, St. Louis, MO, USA), respectively, for 48 and 24 h before treatments to obtain more differentiated cell lines.

### 4.7. Cell Viability Assay

To evaluate cell viability, Saos-2 and MG63 cells were seeded at a density of 1 × 10^4^ cells/well in 96-well plates. Cells were treated in serum-free medium and incubated with ZSEs at concentrations ranging from 0.15 to 40 µg/mL for 24 h. Cell viability was determined using the 3-(4,5-dimethylthiazol-2- yl)-2,5-diphenyltetrazolium bromide (MTT) assay. Briefly, 10 µL of MTT solution (5 mg/mL) was added to each well and incubated at 37 °C for 2 h. The supernatants were then removed and replaced with 100 µL of DMSO. The absorbance of the obtained solution was measured at a wavelength of 595 nm by a multiplate spectrophotometer (BioTek–800TS microplate reader, Agilent, Santa Clara, CA, USA).

### 4.8. Western Blotting

To investigate the pathways involved in the bone remodeling process, Saos-2 and MG63 cells were seeded at a density of 4 × 10^5^ cells/well in p60 plates. Cultures were starved with serum-free media for 4 h before treatments. Cells were treated in serum-free medium and incubated with ZSEs at concentration ranging from 2.5 to 40 µg/mL for 10 min. Cells were lysed in Mammalian Protein Extraction Reagent (M-PER) (Pierce, Thermo Fisher Scientific). The medium from the cell cultures was centrifuged using Amicon Ultra Centrifugal Filters (Merck Millipore, Milano, Italy) to concentrate the extracellular proteins. The total amount of proteins was quantified using the Bradford spectrophotometric assay. A Western blot analysis of the proteins extracted from the cell lysates was performed according to standard procedures. The following antibodies were used: rabbit anti-p Extracellular Signal-regulated Kinase (ERK)1/2 (9101), and mouse anti-pAKT(3700), by Cell Signaling Technology (Beverly, MA, USA); from Sigma Aldrich (St. Louis, MO, USA); ß-actin by Abcam (Cambridge, UK).

### 4.9. Kit ROS DCFDA

The cellular ROS Assay Kit [AB113851] DCFDA/H2DCFDA was used as prescribed by the supplier company. In detail, bone cells (2.5 × 10^4^) were seed on a 96-wells plate and allowed to attach overnight.

The culture media were removed, and cells were washed with 1x buffer provided. Subsequently, cells were stained with DCFDA 25 µM in 1x buffer and incubated for 45 min at 37 °C. Cells were washed again with the 1x buffer and 100 µL of the solution containing the ZSEs was added to each well. After 45 min incubation at 37 °C, the 96-well plate were analyzed using Gloomax Discover (Promega-Promega Italia Srl-Milano). In detail, each well was exited at wavelength 475 nm and signals emitted between 500 and 550 nm were captured. Background noise was subtracted from all signals obtained and the results were expressed as the percentage variation in comparison with the control group.

### 4.10. Statistical Analysis

All data describe the average of three independent experiments and are expressed as the mean including distribution spots for standard deviation or percentage in weight and volume. Data concerning all physicochemical properties of the extracts were processed using Microsoft^®^ Excel^TM^ (Microsoft office 2016). HPLC and UV/VIS graphs were realized with SigmaPlot 12.0 (Systat Software INC, 2011, Chicago, IL, USA). All data related to the in vitro study were analyzed with the GraphPad Prism 5.0 software using a two-tailed Student’s t-test for pairwise comparisons and an ANOVA test to compare the means among three or more groups. Significant differences were assumed to be present at *p* < 0.05. The * symbol was used to compare the data with control group, while ^Ψ^ describes statistical significance versus the solvent used for the extracts (ethanol 70% *v*/*v*). In detail, we used * and ^Ψ^ for *p* < 0.05; ** and ^Ψ Ψ^ for *p* < 0.01; *** and ^Ψ Ψ Ψ^ *p* < 0.001 and **** *p* < 0.0001.

## 5. Conclusions

Seeds are an important food source of biomolecules. Zibibbo grape seeds were shown to contain a significant number of phenolic compounds, especially flavonoids, which have strong antioxidant and free radical scavenging properties. Polyphenolic extracts from Zibibbo grape seeds (ZSEs) have been shown to exert beneficial effects on bone metabolism and promote osteoblast proliferation, thus probably helping to counteract and reduce osteodegenerative diseases, such as osteoporosis. The consumption of Zibibbo grape seeds could, therefore, help mitigate the aging process and delay osteodegenerative processes. If confirmed by further analysis, the phenolic compounds in Zibibbo grape seeds could represent a valid adjuvant therapy for treating osteoporosis.

## 6. Limitations of the Study

Despite the promising results achieved in this work, this study has some limitations. First, the effects potentially deriving from lipophilic molecules usually contained in the seeds were not evaluated. Additionally, more in-depth chemical analyses based on mass spectrometry (LC/MS) could provide a more accurate profile of the extracts. Furthermore, other foods rich in flavonoids, especially rutin and hesperidin, were not considered and compared. Finally, other specific pathways involved in bone metabolism could be evaluated to validate and strengthen the results of this study.

## Figures and Tables

**Figure 1 pharmaceuticals-17-01418-f001:**
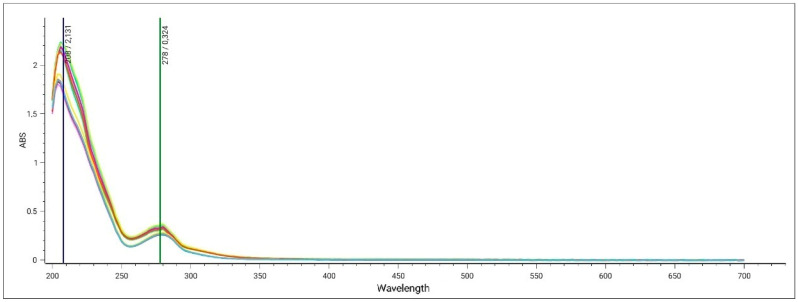
Spectrophotometric analyses of Zibibbo seed extracts (ZSEs) at wavelengths (λ) between 200 nm and 700 nm. Different colors refer to the analysis of different extractions.

**Figure 2 pharmaceuticals-17-01418-f002:**
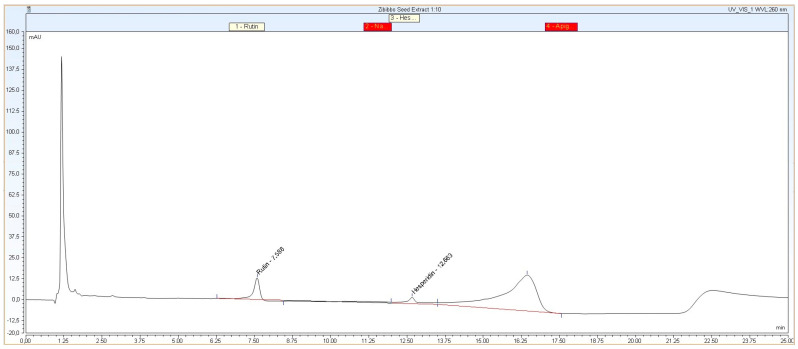
Chromatogram of Zibibbo seed extracts (ZSEs) analyzed by HPLC equipped with a 100 mm reverse-phase C18 column. Acquisition wavelength (λ) was set at 260 nm.

**Figure 3 pharmaceuticals-17-01418-f003:**
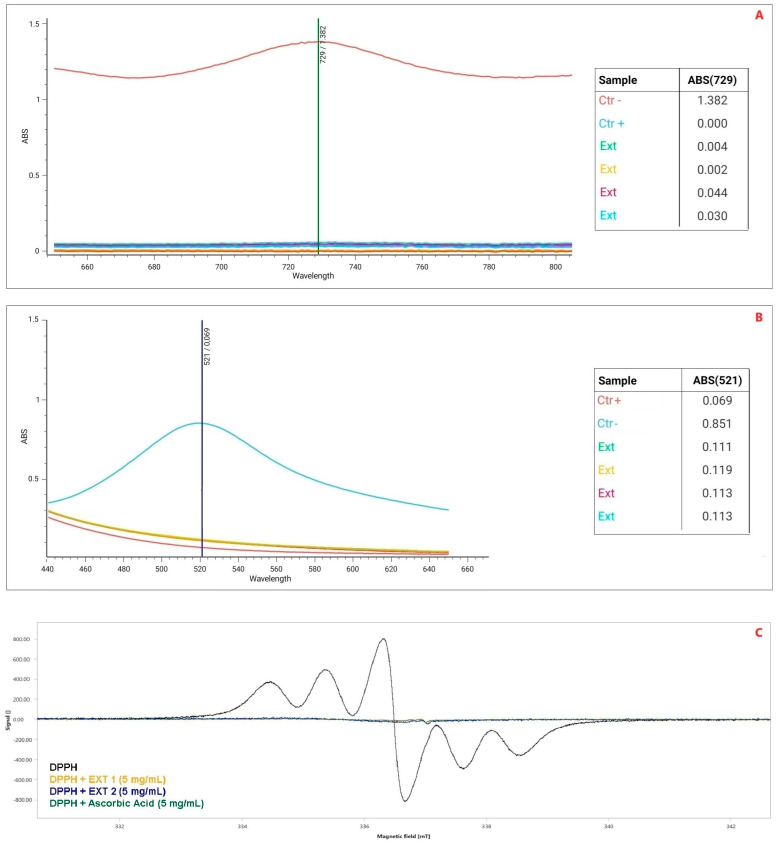
Antioxidant activity of Zibibbo seed extracts (ZSEs) analyzed by ABTS assay (**A**), DPPH assay (**B**) and through Electronic Paramagnetic Resonance (EPR-**C**). CTR+ = positive control; CTR− = negative control; DPPH = 2,2-Diphenyl-1-picrylhydrazyl; EXT = extract.

**Figure 4 pharmaceuticals-17-01418-f004:**
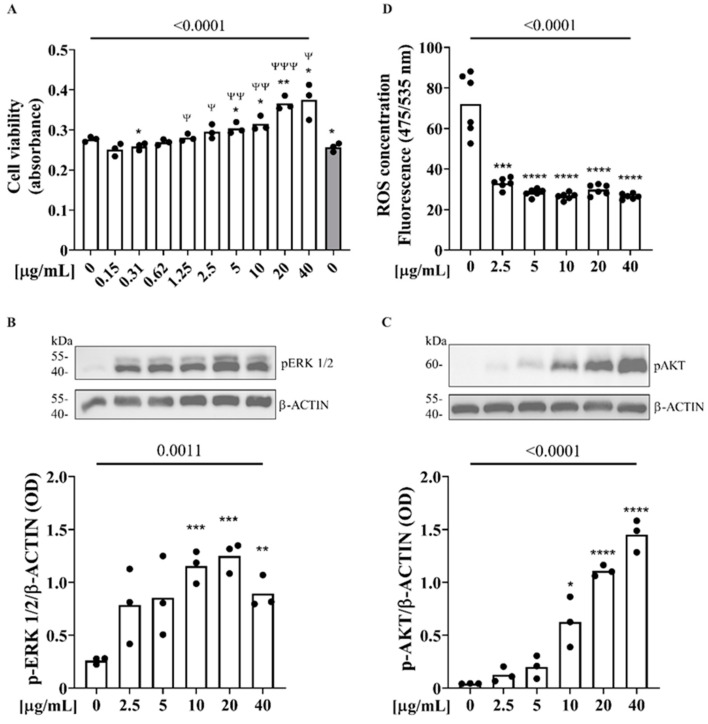
Effect of ZSEs on Saos-2 osteoblast cells’ proliferation after 24 hours of treatment (**A**). Variation in ERK1/2 and AKT protein phosphorylation level after 10 min treatment with ZSE ((**B**,**C**), respectively). Cell proteins were analyzed by Western blotting using specific antibodies to detect phosphorylated ERK and AKT, as well as for ß-actin. Evaluation of radical oxygen species released by Saos-2 cells after 45 min treatment with ZSE, according to the ROS Assay Kit [AB113851] (**D**). Data are expressed as mean of independent experiments (three at least) including distribution spots for standard deviation. (* and ^Ψ^ *p* < 0.05; ** and ^ΨΨ^ *p* < 0.01; *** and ^ΨΨΨ^ *p* < 0.001; **** *p* < 0.0001).

**Figure 5 pharmaceuticals-17-01418-f005:**
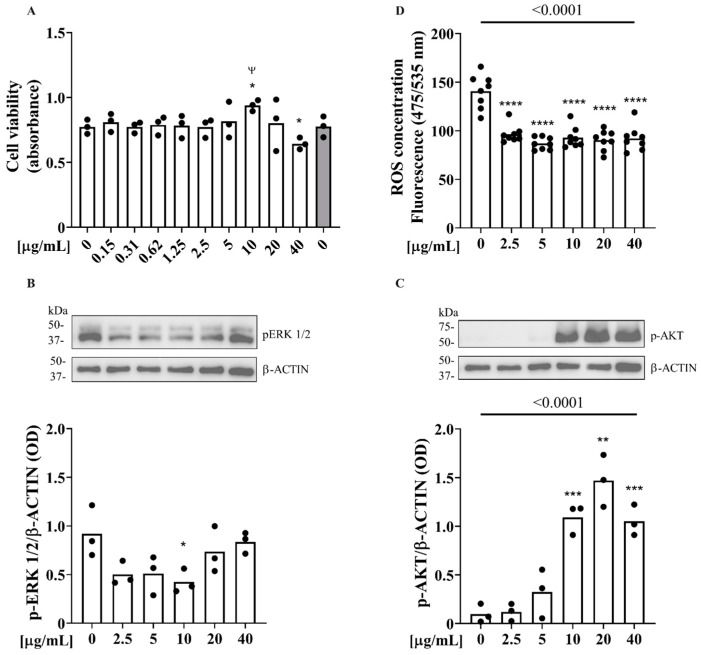
Effect of ZSEs on MG63 osteoblast cells’ proliferation after 24 treatment (**A**). Variation in ERK1/2 and AKT protein phosphorylation level after 10 min treatment with ZSEs ((**B**,**C**), respectively). Cell proteins were analyzed by Western blotting using specific antibodies to quantify phosphorylated ERK and AKT, as well as for ß-actin. Evaluation of radical oxygen species released by MG63 cells after 45 min treatment with ZSEs, according to the ROS Assay Kit [AB113851] (**D**). Data are expressed as mean of independent experiments (three at least) including distribution spots for standard deviation (* and ^Ψ^ *p* < 0.05; ** *p* < 0.01; *** *p* < 0.001; **** *p* < 0.0001).

**Table 1 pharmaceuticals-17-01418-t001:** Chemical characterization of Zibibbo seeds extracts (ZSEa) using a UV-Vis spectrophotometer.

Macromolecule	Method	Amount ± SD
Proteins	Bradford Assay	~0.69 ± 0.02 mg/mL EBSA ^1^
Carbohydrates	Phenol–Sulfuric Acid	~3.23 ± 0.49 mg/mL SEq ^2^
Phenolic Content	Folin–Ciocâlteu Assay	~1.575 ± 0.19 mg/mL GAE ^3^
Flavonoids	Nitrite–Aluminum Assay	~ 2.05 ± 0.25 mg/mL ^4^

^1^ EBSA = equivalent to bovine serum albumin; ^2^ SEq = equivalent to sucrose; ^3^ GAE = equivalent to gallic acid; ^4^ quantified using a flavonoids mixture calibration curve. Data are expressed as mean ± standard deviation (SD).

## Data Availability

The original contributions presented in the study are included in the article/Appendix A, further inquiries can be directed to the corresponding author.

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
