# Peer review of "Zibibbo Grape Seeds’ Polyphenolic Profile: Effects on Bone Turnover and Metabolism"

_pharmaceuticals, 2024, doi:10.3390/ph17111418_

Round 1
Reviewer 1 Report
Comments and Suggestions for Authors
Reviewing the paper entitled as : Zibibbo grape seeds polyphenolic profile: effects on bone turnover and metabolism.
The manuscript is interesting to some extent and is well written. Some English check should be carried out.
However i have some concerns about evidences of some points.
1- The authors focused only on the quantitative determination of flavonoids and some polyphenolic compounds as total fraction while normally the grape extract are very rich in anthocyanins which may haver more poerful role in bone turnover that the suggested and quantified compounds.
2- The authors mentioned that they isolated rutin and hesperidin so what is the source of other used flavonoids(it is not mentioned in the material and methods)
3-I think that LC/Ms techniques could give more accurate results about the real content of other phenolics that may be responsible for the bone turnover why the authors focused only on these flavonoids only.
4- I think that the authors should add more references about the bone turn over effect of the major anthocyanins present in almost all grapes types in the introduction.
Comments on the Quality of English Language
Reviewing the paper entitled as : Zibibbo grape seeds polyphenolic profile: effects on bone turnover and metabolism.
The manuscript is interesting to some extent and is well written. Some English check should be carried out.
Author Response
REVIEWER #1
Reviewing the paper entitled as : Zibibbo grape seeds polyphenolic profile: effects on bone turnover and metabolism.
The manuscript is interesting to some extent and is well written.
Some English check should be carried out.
RESPONSE: We thank the reviewer for this suggestion. After inserting all correction in the manuscript, the draft was checked by ProWritingAid (ProWritingAid - © 2024 - Orpheus Technology, prowritingaid.com) a English lexical editing service offered by our University Library System. Corrections are visible in red in the text.
However i have some concerns about evidences of some points.
1- The authors focused only on the quantitative determination of flavonoids and some polyphenolic compounds as total fraction while normally the grape extract are very rich in anthocyanins which may have more powerful role in bone turnover that the suggested and quantified compounds.
RESPONSE: We thank the reviewer for the opportunity to clarify this aspect. We are aware that grapes are abundant in anthocyanins, whose analysis could be used in future studies. However, several studies demonstrated that the physical-chemical composition of leaves, fruits and seeds belonging to the same plant species significantly differs in qualitative and quantitative terms. For this reason, we analyzed the molecules contained in the seeds of the Zibibbo grape in a completely unconditioned way. We have better clarified this concept in the discussion section with suitable references (lines 263 and 272 and references 21 and 22).
- Zarev, Y.; Marinov, L.; Momekova, D.; Ionkova, I. Exploring phytochemical composition and in vivo anti-inflammatory potential of grape seed oil from an alternative source after traditional fermentation processes: Implications for phytotherapy. Plants 2023, 12, 2795.
- Dussert, S.; Guerin, C.; Andersson, M.; Joët, T.; Tranbarger, T.J.; Pizot, M.; Sarah, G.; Omore, A.; Durand-Gasselin, T.; Morcillo, F. Comparative transcriptome analysis of three oil palm fruit and seed tissues that differ in oil content and fatty acid composition. Plant physiology 2013, 162, 1337-1358.
2- The authors mentioned that they isolated rutin and hesperidin so what is the source of other used flavonoids(it is not mentioned in the material and methods).
ANSWER: Thank you for your question. Rutin, naringin, apigenin and hesperidin were chosen as representative molecules of the different categories of flavonoids due to their structural diversity. However, only rutin and hesperidin were actually present in the extract. We explained this aspect in more detail in the materials and methods section (lines 455 and 460).
3-I think that LC/Ms techniques could give more accurate results about the real content of other phenolics that may be responsible for the bone turnover why the authors focused only on these flavonoids only.
RESPONSE: Thank you for this suggestion. We agree that some techniques such as LC/MS could better characterize the extract. However, due to differences in mass spectroscopy techniques, a simple type of LC/MS analysis would not be sufficient to provide a complete characterization of the extracts, especially considering the heterogeneity of the molecules contained. For this specific reason, several studies exclusively rely on colorimetric spectrophotometric analyses able to provide approximate information about the extracts composition (ref. 26 and 27). In this case, we specifically added HPLC data to provide a more in-depth analysis spectrum. Nevertheless, in agreement with your request, we better clarified this aspect in discussion section (lines 290-299). We also added in the study limits the sentence: “Additionally, more in-depth chemical analyses based on mass spectrometry (LC/MS) could provide a more accurate profile of the extracts” (lines 548 and 550).
Ref 26. - Ispiryan, A.; Atkociuniene, V.; Makstutiene, N.; Sarkinas, A.; Salaseviciene, A.; Urbonaviciene, D.; Viskelis, J.; Pakeltiene, R.; Raudone, L. Correlation between Antimicrobial Activity Values and Total Phenolic Content/Antioxidant Activity in Rubus idaeus L. Plants 2024, 13, 504.
Ref 27. - Du, J.; Li, X.; Liu, N.; Wang, Y.; Li, Y.; Jia, Y.; An, X.; Qi, J. Improving the quality of Glycyrrhiza stems and leaves through solid-state fermentation: flavonoid content, antioxidant activity, metabolic profile, and release mechanism. Chemical and Biological Technologies in Agriculture 2024, 11, 105.
4- I think that the authors should add more references about the bone turn over effect of the major anthocyanins present in almost all grapes types in the introduction.
RESPONSE: In agreement with your request, we completely revised the introduction, aiming to underline the study outcomes. We additionally added in discussion section the sentence: “This study analyzes for the first time the effect of flavonoids contained in Zibibbo grape seeds, finally suggesting their direct involvement in bone turnover. These represent the major advantages and innovations of the re-sults obtained, especially considering that most of the studies in the literature are exclusively based on molecules contained in fruits and pulp, such as an-thocyanins [34, 35].” (lines 332-337). We also added references 34 and 35.
REF 34. Quek, Y.Y.; Cheng, L.J.; Ng, Y.X.; Hey, H.W.D.; Wu, X.V. Effectiveness of anthocyanin-rich foods on bone remodeling biomarkers of middle-aged and older adults at risk of osteoporosis: a systematic review, meta-analysis, and meta-regression. Nutrition Reviews 2024, 82, 1187-1207.
REF 35. Novita, N.; Pudyani, P.S.; Alhasyimi, A.A.; Suparwitri, S. Effect of anthocyanin to post-orthodontic treatment: a review. Indonesian Journal of Orthodontics (InJO) 2024, 1, 1-8.
Reviewer 2 Report
Comments and Suggestions for Authors
Settino and colleagues report Zibibbo grape seeds polyphenolic profile and their effects on bone turn- over and metabolism. The manuscript is interesting. However, despite these findings would be of general interest to this field of research, some points need to be addressed.
Major points
- The statistical analysis is not described. Only the p values are reported but it is not easy to understand the statistical approaches. The Authors must describe the statistical approaches used including all the statistical values obtained.
- The rationale for conducting this study must be better discussed. In particular, the Authors must provide more references regarding the positive impact of diet enriched of polyphenols on health (DOI: 10.3390/antiox12020272; DOI: 10.1093/ajcn/81.1.215S).
- The discussion is very short. The Authors must better discuss the results obtained according to the available literature of the field. The translational/medical aspect should be emphasized.
Minor points
- The Authors should check for typos throughout the manuscript.
- The Authors should also check for the presence of statements without references throughout the manuscript.
Comments on the Quality of English Language
Minor editing
Author Response
REVIEWER #2
Settino and colleagues report Zibibbo grape seeds polyphenolic profile and their effects on bone turn- over and metabolism. The manuscript is interesting. However, despite these findings would be of general interest to this field of research, some points need to be addressed.
Major points
- The statistical analysis is not described. Only the p values are reported but it is not easy to understand the statistical approaches. The Authors must describe the statistical approaches used including all the statistical values obtained.
RESPONSE: We thank the reviewer for this suggestion. This was a typo due to the insertion of the text in the journal template. We have included the statistical analysis in the methods (section 4.9 – line 521-533).
- The rationale for conducting this study must be better discussed. In particular, the Authors must provide more references regarding the positive impact of diet enriched of polyphenols on health (DOI: 10.3390/antiox12020272; DOI: 10.1093/ajcn/81.1.215S).
RESPONSE: We thank the reviewer for this suggestion. In agreement with your request, we added the suggested references and the sentence: “Adherence to the Mediterranean diet appears to be the most beneficial approach to pursuing healthy aging. Most of the recognized benefits of the Mediterranean diet are related to the natural content of foods in polyunsaturated fatty acids, vitamins and polyphenolic compounds. In particular, polyphenols demonstrated to play a key role in promoting people's health and the prevention of various diseases, such as type 2 diabetes, cardiovascular diseases, obesity and even neurological pathologies. In fact, polyphenols are known to be directly involved in glucose regulation and transportation, as well as in protection of β-cells from glucose toxicity [15,16]. They also exerted effects on reduction of platelet activation and reduction of LDL cholesterol [17], on adipocyte oxidation [18] and on psychological and cognitive health, cases in which the consumption of coffee, tea, and red wine resulted in a reduction of depressive symptoms and perceived stress [19]”. (lines 242-254)
- Prasad, C.; Davis, K.E.; Imrhan, V.; Juma, S.; Vijayagopal, P. Advanced glycation end products and risks for chronic diseases: intervening through lifestyle modification. American journal of lifestyle medicine 2019, 13, 384-404.
- Addepalli, V.; Suryavanshi, S.V. Catechin attenuates diabetic autonomic neuropathy in streptozotocin induced diabetic rats. Biomedicine & Pharmacotherapy 2018, 108, 1517-1523.
- Vaisman, N.; Niv, E. Daily consumption of red grape cell powder in a dietary dose improves cardiovascular parameters: A double blind, placebo-controlled, randomized study. International journal of food sciences and nutrition 2015, 66, 342-349.
- Boccellino, M.; D’Angelo, S. Anti-obesity effects of polyphenol intake: Current status and future possibilities. International journal of molecular sciences 2020, 21, 5642.
- Micek, A.; Jurek, J.; Owczarek, M.; Guerrera, I.; Torrisi, S.A.; Castellano, S.; Grosso, G.; Alshatwi, A.A.; Godos, J. Polyphenol-rich beverages and mental health outcomes. Antioxidants 2023, 12, 272.
- The discussion is very short. The Authors must better discuss the results obtained according to the available literature of the field. The translational/medical aspect should be emphasized.
RESPONSE: Many thanks for this suggestion. In agreement with your request, we increased the discussion section, also emphasizing the advantages related to the obtained results in comparison to previous published data.
Minor points
- The Authors should check for typos throughout the manuscript.
RESPONSE: We thank the reviewer for this suggestion. After inserting all correction in the manuscript the draft was checked by ProWritingAid (ProWritingAid - © 2024 - Orpheus Technology, prowritingaid.com) a English lexical editing service offered by our University Library System. Corrections are visible in red in the text.
- The Authors should also check for the presence of statements without references throughout the manuscript.
RESPONSE: We checked the manuscript and inserted suitable references for data described. The complete revision allowed to add 13 new references.

Reviewer 3 Report
Comments and Suggestions for Authors
Overall, the manuscript is well written, but some aspects need improvement. The manuscript has significant weaknesses in justification, methods, results, and discussion, which prevent its recommendation for publication in its current form.
Introduction. The study lacks sufficient justification. What makes conducting a study of this variety particularly interesting or special? Or what knowledge gap does the study aim to address?
Material and methods. To replicate the work it is necessary to know the origin of the grape seeds. From what area? How many populations were evaluated? Under what conditions do these populations grow? Climate? Soils? Etc.
Results. The number of molecules identified is little concerning the number of peaks obtained. The justification for the lack of standards is not supported given that there are works on grape seeds where a greater number of compounds are identified. It is necessary to consult them and compare their results, if there are differences, how are they explained?
Discussion. Due to the mentioned problems, the discussion must change, as it is, is too general, and does not highlight the importance of the results found, comparatively with other works where grape seeds are characterized.
Consult Ma et Zhang, 2017; Zarev et al. 2023, Karagecili et al. 2023, for example.
Author Response
REVIEWER #3
Overall, the manuscript is well written, but some aspects need improvement. The manuscript has significant weaknesses in justification, methods, results, and discussion, which prevent its recommendation for publication in its current form.
RESPONSE: We thank the reviewer for all the suggestions provided. Appropriate corrections have been made to the manuscript based on the weaknesses highlighted.
Introduction. The study lacks sufficient justification. What makes conducting a study of this variety particularly interesting or special? Or what knowledge gap does the study aim to address?
RESPONSE: We changed the introduction section and improved its complete content (lines 49-99).
Material and methods. To replicate the work it is necessary to know the origin of the grape seeds. From what area? How many populations were evaluated? Under what conditions do these populations grow? Climate? Soils? Etc.
RESPONSE: In agreement with your request, we changed the 4.2. method section and improved its content including more details about the grapes (lines 355-363).
Results. The number of molecules identified is little concerning the number of peaks obtained. The justification for the lack of standards is not supported given that there are works on grape seeds where a greater number of compounds are identified. It is necessary to consult them and compare their results, if there are differences, how are they explained?
RESPONSE: We thank the reviewer for the opportunity to clarify this aspect. We better described the HPLC analyses (lines 133 – 144) and we discussed the data obtained in discussion section (lines 290-299). Finally, we also added in limits of the study the sentence: “Additionally, more in-depth chemical analyses based on mass spectrometry (LC/MS) could provide a more accurate profile of the extracts”.
Discussion. Due to the mentioned problems, the discussion must change, as it is, is too general, and does not highlight the importance of the results found, comparatively with other works where grape seeds are characterized.
Consult Ma et Zhang, 2017; Zarev et al. 2023, Karagecili et al. 2023, for example.
RESPONSE: In agreement with your request, we reviewed all sections of the manuscript, finally increasing the citations and references, as well as the discussion, in order to highlight the importance of data obtained in this study.

Round 2
Reviewer 1 Report
Comments and Suggestions for Authors
Thanks for adding the required information and references.
Reviewer 2 Report
Comments and Suggestions for Authors
The Authors have addressed all the points I raised.
Comments on the Quality of English LanguageMinor editing
Reviewer 3 Report
Comments and Suggestions for Authors
The authors adequately addressed the suggestions made in the first revision of the manuscript.